# Axial Attention
# In Multidimensional Transformers

## Abstract

We propose Axial Transformers, a self-attention-based autoregressive model for images and other data organized as high dimensional tensors. Existing autoregressive models either suffer from excessively large computational resource requirements for high dimensional data, or make compromises in terms of distribution expressiveness or ease of implementation in order to decrease resource requirements. Our architecture, by contrast, maintains both full expressiveness over joint distributions over data and ease of implementation with standard deep learning frameworks, while requiring reasonable memory and computation and achieving state-of-the-art results on standard generative modeling benchmarks. Our models are based on axial attention, a simple generalization of self-attention that naturally aligns with the multiple dimensions of the tensors in both the encoding and the decoding settings. Notably the proposed structure of the layers allows for the vast majority of the context to be computed in parallel during decoding without introducing any independence assumptions. This semi-parallel structure goes a long way to making decoding from even a very large Axial Transformer broadly applicable. We demonstrate state-of-the-art results for the Axial Transformer on the ImageNet-32 and ImageNet-64 image benchmarks as well as on the BAIR Robotic Pushing video benchmark. We open source the implementation of Axial Transformers.

## 1 Introduction

Autoregressive models are a family of exact likelihood-based generative models that represent the joint distribution of data $x = (x_1, \ldots, x_N)$ as a product of conditionals $p_\theta(x) = \prod_{i=1}^{N} p_\theta(x_i \mid x_{<i})$. Neural network models in this family have achieved state-of-the-art log likelihoods on high-dimensional image and video datasets (van den Oord et al., 2016a; Chen et al., 2018; Menick & Kalchbrenner, 2018; Parmar et al., 2018; Child et al., 2019; Weissenborn et al., 2019; Salimans et al., 2017; Kalchbrenner et al., 2017; Uria et al., 2016; Parikh et al., 2016; Theis & Bethge, 2015; van den Oord et al., 2016b) due to architectural innovations that enable the following capabilities:

1. Large, high information bandwidth receptive fields for each pixel $x_i$, capable of expressing long-range dependencies over previous pixels $x_{<i}$, and

2. Computationally efficient, vectorizable computation of the log likelihood and its gradient.

Autoregressive model architectures that can read long-range dependencies over large receptive fields are able to express all joint distributions over the data. Meanwhile, architectures that admit fast log likelihood gradient computation are suitable for training using a stochastic gradient method on a maximum likelihood objective—a straightforward, stable training procedure for generative models.

These desiderata make self-attention a compelling building block for autoregressive model architectures. Self-attention is a neural network operation that is able to transform a sequence $y_1, \ldots, y_N$ into a sequence $y'_1, \ldots, y'_N$, where each $y'_i$ depends on all $y_i$ by way of a single vectorizable computation (Vaswani et al., 2017). Self-attention is remarkably effective at learning long-range dependencies between data dimensions and neural networks that incorporate self-attention in their designs are state-of-the-art on many tasks from language modelling and machine translation to image and video modelling (Parmar et al., 2018; Child et al., 2019).

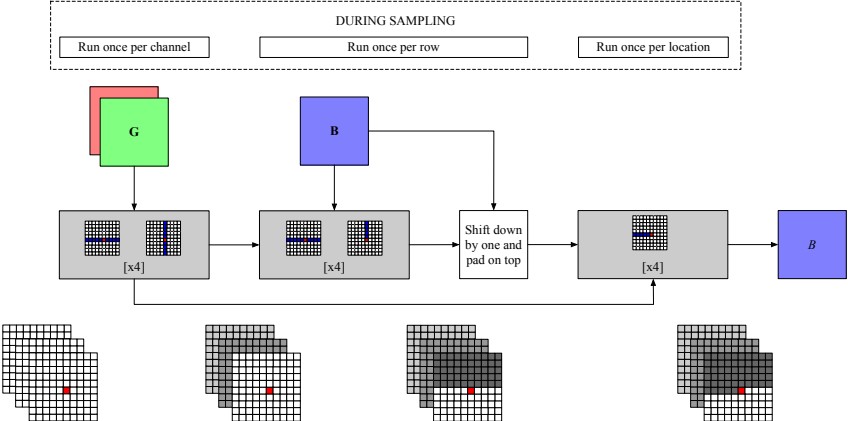

Figure 1: The Axial Transformer model for 2-dimensional tensors. Before sampling a channel we encode all previous channels and frames with 8 blocks of unmasked row and unmasked column attention (left). Then, for each row, we apply 4 blocks of unmasked row and masked column attention to integrate the previously sampled rows for the active channels into our encoded representation (middle). Finally, we shift the encoded representation up to make sure the conditioning information satisfies causality, and we run the inner decoder consisting of 4 blocks of masked row attention to sample a new row in the image (right).

But the power of self-attention comes at the price of computational complexity. The memory and computation it consumes grow quadratically with the sequence length $N$ making it prohibitively expensive to directly apply self-attention to long sequences. In the case of autoregressive models of multidimensional tensors such as images or videos, the aim to capture large receptive fields in multiple dimensions further exacerbates the problem as even a modest number of receptive field steps in each dimension can encompass a large total number of locations. Various approaches have been proposed to alleviate this difficulty at the cost of either limiting the receptive field or requiring operations that may not be broadly available on GPUs or TPUs.

We propose the Axial Transformer, a simple yet effective self-attention-based autoregressive model for data organized as multidimensional tensors. Rather than applying attention to a flattened string of tensor elements, our model instead applies attention along a single axis of the tensor without flattening—we refer to this as "axial attention." Since the length of any single axis (that is, the height or width of an image) is typically much smaller than the total number of elements, an axial attention operation enjoys a significant saving in computation and memory over standard self-attention: for a $d$-dimensional tensor with shape $N = N^{1/d} \times \cdots \times N^{1/d}$, axial attention saves a $O(N^{(d-1)/d})$ factor of resources over standard self-attention.

Our Axial Transformer architecture allows for the majority of the context $x_{<i}$ to be embedded with a high degree of parallelism *without* introducing conditional independence assumptions among any of the locations, but has an interesting property that it is amenable to a simple-to-implement fast sampling procedure. To sample one row of an image, the Axial Transformer only runs an autoregressive Transformer over that one row only, without re-embedding pixels from previous rows. We structure the Axial Transformer, however, so that it always defines a fully expressive joint distribution. No dependencies on previous pixels are ever lost.

We evaluate Axial Transformers on image and video modelling benchmarks. We show that Axial Transformer achieves state-of-the-art results on ImageNet-32 and on ImageNet-64. We also show that, simply by stacking a video along the channel dimension, the Axial Transformer can be directly applied to the channel-stacked video without nearly any modification. On the BAIR Robot Pushing benchmark, the Axial Transformer significantly outperforms previous results without using an architecture specially designed for videos. The generated samples on these datasets are of the expected high quality.

Axial Transformers do not require subroutines for GPUs or TPUs that may exhibit unfavorable memory bandwidth and computation trade-offs. Axial Transformers are simple to implement using

Figure 2: Types of axial attention layers that are the building blocks of the Axial Transformer. The blue locations correspond to the receptive field of the output red location.

efficient operations that are widely available in deep learning frameworks (primarily dense-dense MatMuls). An open source implementation of our models is available at `anonymized URL`.

## 2 BACKGROUND

To set the stage for our discussion, we first review self-attention and its computational resource requirements in the context of autoregressive modeling. A self-attention layer takes as input a length $N$ sequence of $D$-dimensional embeddings $X$ (a $N \times D$ matrix) and produces an output sequence $Y$ (also a $N \times D$ matrix) via:

$$Q = XW_Q, \quad K = XW_K, \quad V = XW_V$$
$$A = \text{softmax}\left(QK^\top/\sqrt{D}\right), \quad Y = AV$$

$W_Q$, $W_K$, and $W_V$ are $D \times D$ parameter matrices responsible for projecting the entries of the sequence $X$ into keys, queries, and values, respectively. Each entry of the output sequence $Y$ is a linear combination of values in $V$ weighted by the attention matrix $A$, which itself is computed from similarities between all pairs of query and key vectors. Both the expressive power and the resource cost of self-attention come from computing $A$ and $Y$: it takes $O(N^2)$ time and space to compute the pairwise similarities between $Q$ and $K$ and to compute the linear combination of $V$ vectors.

This quadratic complexity makes it impractical to apply self-attention to images and videos directly as flattened vectors: a small $32 \times 32 \times 3$ image has 3072 dimensions. Sequences such as these are too long for self-attention, so attempts to scale self-attention to these modalities generally involve restricting these sequence lengths in a modality-aware manner while attempting to preserve modeling performance.

One strategy is to restrict the conditioning context $x_{<i}$ to a carefully designed small subset of the data dimensions. While this reduces the cost of attention, which is only performed over these small subsets instead of the full data, the model can no longer express all joint distributions over the data. Parmar et al. (2018) propose image models with conditioning context $x_{<i}$ restricted to a small window of the full image, but the implementation requires redundant data copies to extract and process these windows. Weissenborn et al. (2019) similarly scale video autoregressive models by restricting the context, again preventing their model from expressing all joint distributions over pixels. Our models do not restrict context and hence we obtain better log likelihoods, as we will see in section 4.

A different strategy is to stack multiple sparse attention layers, each with restricted context for computational efficiency, but in a manner that overlapping these layers yields a full-context model. Child et al. (2019) propose two sparse attention patterns with this property. However, the architecture they propose that works best for images (the Strided Sparse Transformer) requires custom sparse attention GPU kernels to implement a specific block-sparse variant of matrix-matrix-multiply. The model cannot be easily implemented on other hardware such as TPUs.

See table 1 for a summary of these architecture design tradeoffs. Our goal in this paper is to design attention-based autoregressive models that attain the best of all worlds. Our Axial Transformer, described in subsequent sections, has a full conditioning context, so its ability to express joint distributions is never limited. The Axial Transformer also does not require any redundant data copies or custom kernels to implement in an efficient way. Indeed, we designed, and will make open source, an efficient implementation that uses only standard operations in deep learning libraries.

Table 1: Trade-offs of recently proposed multidimensional Transformer architectures.

| Model | Full receptive field | Attention faster than $O(N^2)$ | Needs no custom kernels | Semi-parallel context aggregation |
|---|---|---|---|---|
| Transformer (Vaswani et al., 2017) | yes | no | yes | no |
| Image Transformer (Parmar et al., 2018) | no | yes | yes | no |
| Block Transformer (Weissenborn et al., 2019) | no | yes | yes | no |
| Strided Sparse Transformer (Child et al., 2019) | yes | yes | no | no |
| Axial Transformer (ours) | yes | yes | yes | yes |

# 3 AXIAL TRANSFORMERS

We now describe Axial Transformers, our self-attention-based autoregressive models for high-dimensional data tensors. We describe its basic building block in section 3.1 and then we complete the description into a full autoregressive model in section 3.2.

## 3.1 AXIAL ATTENTION

We first introduce our basic building block for developing self-attention-based autoregressive models for high-dimensional data tensors. The proposed approach does not change the original shape of the multidimensional data tensor and performs a masked or unmasked attention over a single axis of the tensor at a time. We call this operation *axial attention*, denoted by $\text{Attention}_k(x)$. It performs attention over axis $k$ of the tensor $x$, mixing information along axis $k$ while keeping information along other axes independent. It is straightforward to implement: axial attention over axis $k$ can be implemented by transposing all axes except $k$ to the batch axis, calling standard attention as a subroutine, then undoing the transpose (an alternative is to use the einsum operation available in most deep learning libraries).

When the data is an image, we call $\text{Attention}_1$ *column attention*, as it mixes information within columns while keeping separate columns independent. We call $\text{Attention}_2$ *row attention* for analogous reasons. Axial attention on a square image of size $N = S \times S$ performs attention on $S$ sequences of length $S$—this is a total of $O(S \cdot S^2) = O(N\sqrt{N})$ computation—an $O(\sqrt{N})$ savings in computation over standard self-attention. In general, for a $d$-dimensional tensor with $N = S^d$, axial attention saves $O(N^{(d-1)/d})$ computation over standard attention. Of course, a single layer of axial attention along some axis $k$ does not have the full receptive field since it covers a single axis, but we will see in section 3.2 that stacking two axial attention layers allows the model to obtain a global receptive field.

It will be important for us to also define $\text{MaskedAttention}_k$ to be the causally masked variant of $\text{Attention}_k$: component $i$ of the result of $\text{MaskedAttention}_k(x)$ along axis $k$ depends on only components $1, \ldots, i$ of $x$ along axis $k$. The receptive fields of these attention patterns, both unmasked and masked, are illustrated in fig. 2. We will use these masked blocks to build our autoregressive model in section 3.2.

Axial attention can be used within standard Transformer layers in a straightforward manner to produce Axial Transformer layers. The basic building blocks are the same as those found in the standard Transformer architecture:

- $\text{LayerNorm}(x)$: layer normalization (Ba et al., 2016), and
- $\text{Dense}_D(x)$: a dense layer operating over the last axis of the input $x$. The letter $D$ denotes the dimension of the output activations. If the input has shape $H \times W \times C$, then this operation is identical to a $1 \times 1$ convolution, and the output has shape $H \times W \times D$.

We use these to define ResNet axial attention blocks operating on tensors of $D$-dimensional embeddings (Vaswani et al., 2017; Child et al., 2019):

- $\text{FeedforwardBlock}(x) = x + \text{Dense}_D(\text{Nonlinearity}(\text{Dense}_{D'}(\text{LayerNorm}(x))))$
- $\text{AttentionBlock}_k(x) = x + \text{Dense}_D(\text{Attention}_k(\text{LayerNorm}(x)))$
- $\text{TransformerBlock}_k(x) = \text{FeedforwardBlock}(\text{AttentionBlock}_k(x))$

$D'$ is chosen to be some constant factor larger than $D$, from 1 to 4 (Vaswani et al., 2017). We also define a MaskedTransformerBlock$_k$ using MaskedAttention$_k$ in place of Attention$_k$.

Operations similar to unmasked axial attention have been proposed in other contexts in computer vision (Huang et al., 2019). Our focus in forthcoming sections is the use of *masked* axial attention and its utility in autoregressive image modeling, which is not explored in these works.

## 3.2 AXIAL TRANSFORMERS

We now describe Axial Transformers, our axial attention-based autoregressive models for images and videos. We will use the axial attention operations described in section 3.1 as building blocks in a multi-layer autoregressive model of the form $p_\theta(x) = \prod_{i=1}^{N} p_\theta(x_i \mid x_{<i})$ following the raster scan ordering of pixels. We will accomplish this by building an autoregressive model over rows (section 3.2.1), then conditioning each row on previous rows (section 3.2.1), then further conditioning on previous channels and frames (section 3.2.2). Decomposing the model in this manner also leads to a simple fast and partly parallel sampling procedure (section 3.2.1).

### 3.2.1 A MODEL FOR SINGLE-CHANNEL IMAGES

We begin with an autoregressive model for a single-channel image $x$ with shape $H \times W$, with each pixel taking an integer value in $[0, 255]$ representing its intensity. As is standard practice with Transformers, pixel intensities are first embedded into a $H \times W \times D$ tensor of $D$-dimensional embeddings, which we call $h$. The architecture's responsibility is to transform $h$ into a $H \times W \times 256$ tensor of logits suitable for classification or sampling. These logits must depend only on previous pixels in the input $x$ along the raster scan ordering to ensure that the architecture defines a valid autoregressive model.

**Inner Decoder: a row-wise model**    Our idea is to begin with masked row attention layers to create a "row-wise" model:

$$h \leftarrow \text{Embed}(x)$$
$$h \leftarrow \text{ShiftRight}(h) + \text{PositionEmbeddings}$$
$$h \leftarrow \text{MaskedTransformerBlock}_2(h) \qquad \times L_{\text{row}}$$

Here, $L_{\text{row}}$ is the number of masked row attention blocks applied to $h$. PositionEmbeddings is a $H \times W \times D$ tensor of position embeddings that inform the attention layers of the position. For parameter efficiency we use "additively factorized" position embeddings, meaning that we parameterize them as a broadcasted sum of $H \times 1 \times D$ embeddings for rows and $1 \times W \times D$ embeddings for columns.

The operation ShiftRight shifts the input right by one pixel, which has the effect of shifting the receptive field left by one pixel. This ensures that the masked row attention layers exclude the current pixel from their receptive field, which is crucial for architecture to define a correct autoregressive model.

As this model employs row attention only, it enjoys the computational efficiency benefits described in section 3.1. However, it clearly does not define a full-context model because each location in the output does not depend on input pixels in previous rows. If we were to use the resulting $h$ as logits for pixel intensity prediction, we would obtain a set of $H$ independent autoregressive models $p(x_{i,j}|x_{i,1}, \ldots, x_{i,j-1})$ for each row $i \in [1, H]$, not a single autoregressive model with full context. We address this issue next.

**Outer Decoder: capturing the rows above**    Each pixel $x_{i,j}$ in the aforementioned model already depends on previous pixels in its own row $x_{i,<j}$. We just need to make it depend on all previous rows $x_{<i,:}$ too. So, we insert unmasked row and masked column layers in the beginning of the model as follows (newly inserted operations are underlined):

$$h \leftarrow \text{Embed}(x)$$
$$u \leftarrow \underline{h + \text{PositionEmbeddings}}$$
$$u \leftarrow \underline{\text{MaskedTransformerBlock}_1(\text{TransformerBlock}_2(u))} \qquad \times L_{\text{upper}}/2$$
$$h \leftarrow \underline{\text{ShiftDown}(u)} + \text{ShiftRight}(h) + \text{PositionEmbeddings}$$
$$h \leftarrow \text{MaskedTransformerBlock}_2(h) \qquad \times L_{\text{row}}$$

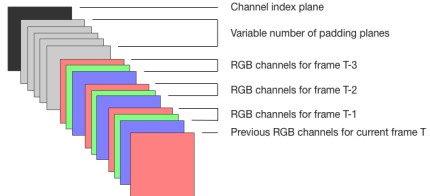

Figure 3: Arrangement of inputs to the encoding network of the Axial Transformer. Previously available or generated channels of an image or video are sequentially stacked in the input. A variable number of padding planes are used as placeholders for future generated channels. A final integer plane signals to the Axial Transformer the channel that is being generated at that step.

The tensor $u$ represents context captured above the current pixel. It is computed by unmasked row and masked column attention layers, repeated to a total of $L_{\mathrm{upper}}$ layers to increase model capacity, which make $u$ cover the receptive field at all rows above and including the current pixel. The ShiftDown operation shifts $u$ down one pixel, which shifts its receptive field up one pixel. Thus we have a context which captures all pixels above while *excluding* the current row, which we add to $h$ as input to the masked row layers. We have thus converted the row-wise model into a fully expressive autoregressive model that captures not only pixels in the current row but also those above.

Following standard practice, we pass the final $h$ through layer normalization and a final dense layer to produce logits with shape $H \times W \times 256$. The logits at each location depend on all previous pixel locations in the raster scan ordering.

**Semi-Parallel Sampling**  Naive implementations of sampling from sequential models are notoriously slow because they require re-evaluating the entire network to sample each location. In the case of our model for a $\sqrt{N} \times \sqrt{N}$ square image, each network evaluation takes $O(N\sqrt{N}(L_{\mathrm{upper}} + L_{\mathrm{row}}))$ time, so sampling the whole image would take $O(N^2\sqrt{N}(L_{\mathrm{upper}} + L_{\mathrm{row}}))$, which is far too large.

Fortunately, our architecture is amenable to a particularly simple implementation of a faster sampling that is able to compute large sections of the model in parallel (see Figure 1). Pseudocode is as follows:

1. For each row $i \in [1, H]$:
    (a) Compute the upper context $u$ including information about all $x_{<i,*}$ using the upper layers
    (b) For each column $j \in [1, W]$:
        i. Sample $x_{i,j}$ conditioned on $u$ and prior elements of row i ($x_{i,<j}$).

Because the $L_{\mathrm{row}}$ row-wise layers are independent over rows (they depend on other rows only through the upper context, as explained in section 3.2.1), sampling one row can be accomplished by evaluating the row-wise layers for that one row only, completely ignoring other rows. Thus, in one row of $\sqrt{N}$ pixels, each pixel can be sampled in $O(NL_{\mathrm{row}})$, so all pixels can be sampled in $O(N^2 L_{\mathrm{row}})$. Before each of the $\sqrt{N}$ rows can be sampled, the upper context must be computed in $O(N\sqrt{N}L_{\mathrm{upper}})$, for a total of $O(N^2 L_{\mathrm{upper}})$ over the course of all rows. Thus we arrive at $O(N^2(L_{\mathrm{upper}} + L_{\mathrm{row}}))$ in total, which is $\sqrt{N}$ faster than the naive implementation. To our knowledge, sampling speedups of this type are not possible with contemporary work on scaling Transformers to images and videos (Child et al., 2019; Weissenborn et al., 2019).

### 3.2.2 CHANNEL ENCODER FOR MULTI-CHANNEL IMAGES AND VIDEOS

We have just described an architecture for a single-channel image of shape $H \times W$. Here, we show how to extend the architecture to multi-channel images or videos of shape $H \times W \times C$ (here $C$ is either the number of channels in a multi-channel image, or the product of the number of channels and timesteps in a video). One way to model such data of shape $H \times W \times C$ is to simply stack the channels on top of each other into a single-channel image of shape $(H \cdot C) \times W$ or $H \times (W \cdot C)$. This

Table 2: Unconditional and class-conditional image modeling results (bits/dim)

| Model | ImageNet 32x32 | ImageNet 64x64 |
|---|---|---|
| Multiscale PixelCNN (Reed et al., 2017) | 3.95 | 3.70 |
| PixelCNN/RNN (van den Oord et al., 2016a) | 3.86 | 3.63 |
| Gated PixelCNN (van den Oord et al., 2016b) | 3.83 | 3.57 |
| PixelSNAIL (Chen et al., 2018) | 3.80 | 3.52 |
| SPN (Menick & Kalchbrenner, 2018) | 3.79 | 3.52 |
| Image Transformer (Parmar et al., 2018) | 3.77 | – |
| Strided Sparse Transformer (Child et al., 2019) | – | **3.44** |
| Axial Transformer + LSTM inner decoder | 3.77 | 3.46 |
| Axial Transformer | **3.76** (3.758) | **3.44** (3.439) |

Table 3: Video modeling results (bits/dim) on the **BAIR Robotic Pushing** dataset (Ebert et al., 2017). We condition on a single video frame and model the next 15 frames, similar to Weissenborn et al. (2019). Kumar et al. (2019) instead condition on the 3 prior frames of the video.

| Model | bits/dim next 15 frames |
|---|---|
| VideoFlow (Kumar et al., 2019) | 1.87 |
| Video Transformer (Weissenborn et al., 2019) | 1.35 |
| Axial Transformer (ours) | **1.29** |

is simple to implement, but does increase the sequence length for column attention or row attention, which can be undesirable for large $C$. We instead opt to model one channel at a time as a single-channel image, but now conditioned on previous channels using an extra set of unmasked row and unmasked column attention layers. This means that we have a model of the form $p(x_{:,:,c} \mid x_{:,:,<c})$, where previous channels $x_{:,:,<c}$ are processed into a $H \times W \times D$ tensor of context information, which is then added into the first encoding blocks of the model in section 3.2.1 (Figure 3).

We do not share any parameters among any of these layers. At training time, we train on a random channel slice of each image: we process the previous slices using these unmasked attention layers to produce a context tensor, and maximize the likelihood of the randomly chosen slice conditioned on this context. This amounts to training on an unbiased estimate of log likelihood for the whole data tensor. See fig. 1 for an illustration of this complete model.

## 4 EXPERIMENTS

We benchmarked our models on standard datasets for generative image and video models: down-sampled ImageNet (van den Oord et al., 2016a) and BAIR Robot Pushing (Ebert et al., 2017). All Axial Transformers have 8 total layers in the encoder, 8 layers in the outer decoder and 4 layers in the inner decoder. We use a hidden size of 2048 neurons throughout and for all setups and 16 heads with 128 neurons each for the attention component. We train for approximately 200k steps on ImageNet32 and ImageNet64 and for 200k steps on BAIR Robot Pushing. Our models can overfit on ImageNet32, but on the other datasets the models keep on gradually improving with more steps. See table 2 and table 3 for our results.

### 4.1 ABLATION STUDY

To push the limits of the semi-parallel sampling by making the inner decoder as small as possible, we train an Axial Transformer with the inner decoder replaced by a single LSTM layer of 2048 units. This slows down training time by about 20% on ImageNet32 and about 80% on ImageNet64 when maintaining the number of steps and all else fixed. We find that the Axial Transformer + LSTM inner decoder performs rather well on the ImageNet32 and ImageNet64 benchmarks (table 2), thereby also showing the effectiveness of the remaining parts of the Axial Transformer that capture the context of the rows above. We also find however that the full four layers of the inner decoder of the Axial

Transformer provide an additional boost in performance as well as significantly faster training. The Axial Transformer + LSTM inner decoder has the advantage of requiring only a couple of matrix-vector products to compute the layers at each autogressive step, comparing favourably with about the 12 matrix-vector products required by the Axial Transformer, but the slower training time would make the LSTM inner decoder quickly impractical for larger tensors.

## 4.2 SAMPLES

In fig. 4 and fig. 5, we show samples from our $64 \times 64$ and $32 \times 32$ ImageNet models. The samples are globally coherent and show visibly recognizable scenes, meaning that our Axial Transformer architecture successfully captures long-range dependencies across thousands of data dimensions in these image datasets. The samples also don't show any architecture-correlated artefacts. In addition, in fig. 6 we show samples from the BAIR Robotic Pushing dataset. The first frame is each row is given by the dataset and the rest are continuation. We note the high quality exactness of details and the very large diversity (at temperature 1.0).

## 5 CONCLUSION

We proposed the Axial Transformer, an self-attention-based autoregressive model for data organized as high dimensional tensors. It is based on axial attention, a simple generalization of self-attention that scales better with the dimension of input data, achieving a $O(N^{(d-1)/d})$ savings in computation and memory for a $d$-dimensional input tensor with $N$ elements. Axial attention is easy to implement and does not require custom kernels to run efficiently on modern accelerators. Axial Transformers use axial self-attention layers and a shift operation to naturally and efficiently build full receptive fields of multidimensional tensors. Our model matches or outperforms the state-of-the-art on ImageNet-32 and ImageNet-64 image benchmarks and sets a significant new state-of-the-art on the BAIR Robot Pushing video benchmark.

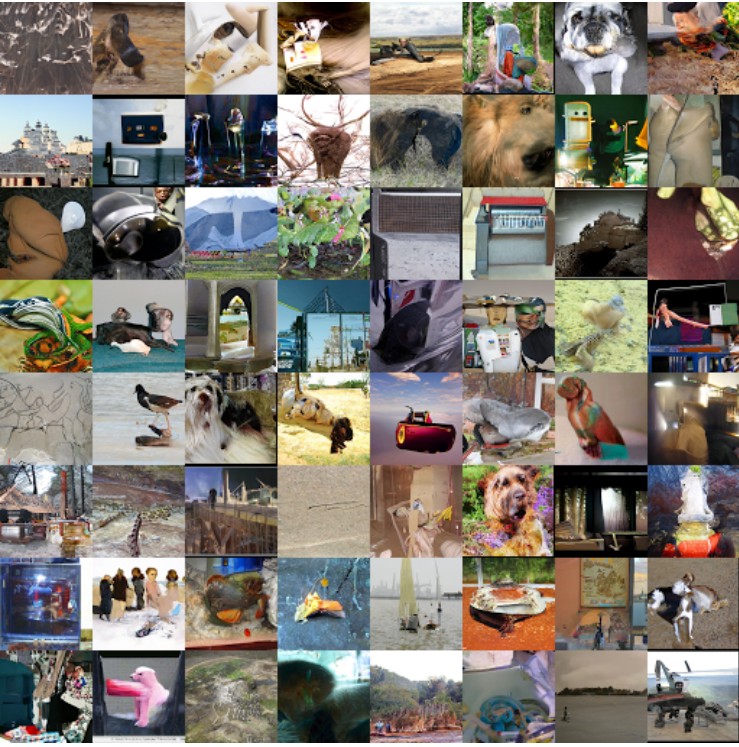

Figure 4: $64 \times 64$ ImageNet samples at temperature 1.0

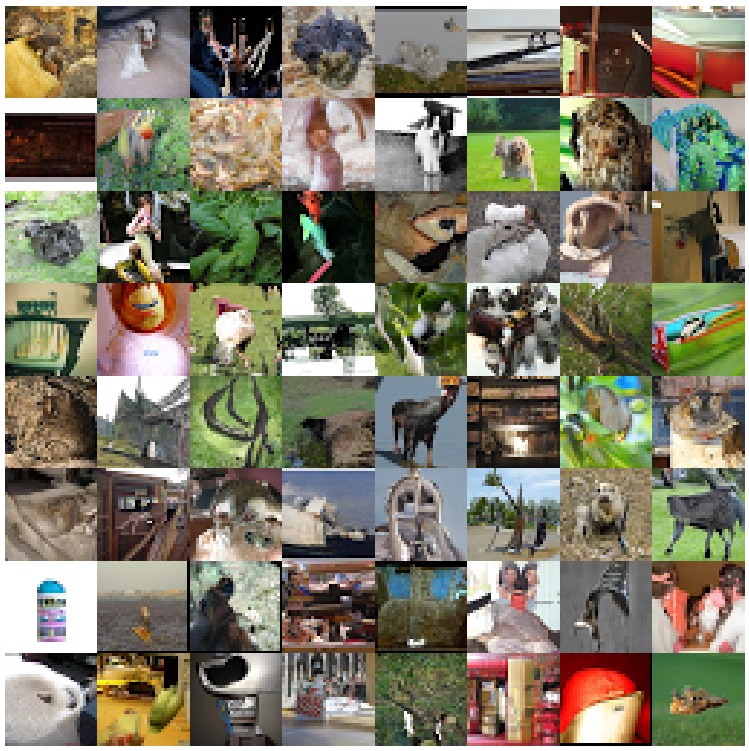

Figure 5: $32 \times 32$ ImageNet samples at temperature 0.99

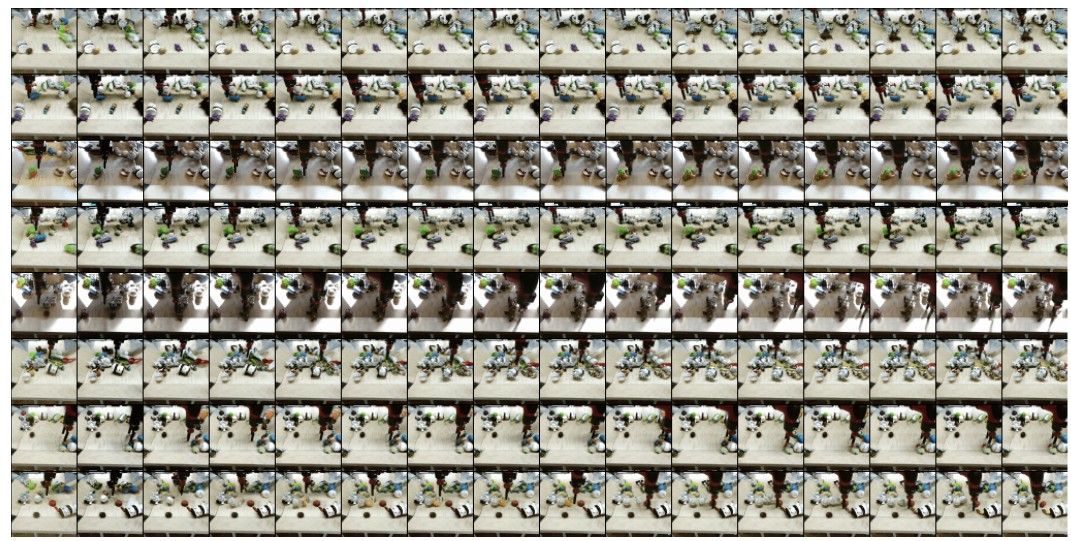

Figure 6: $15 \times 64 \times 64$ BAIR Robot Pushing samples at temperature 1.0

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
