# OpenReview forum: "Axial Attention in Multidimensional Transformers"
_ICLR.cc/2020/Conference — Reject_

### Official Review · AnonReviewer2 · 2019-10-23
**Official Blind Review #2**

**Rating:** 6

**Review:**

This paper proposes axial attention as an alternative of self-attention for data arranged as large multidimensional tensors, which costs too much computational resource since the complexity of traditional self-attention is quadratic in order to capture long-range dependencies for full receptive fields. The axial attention is applied within each axis of the data separately while keeping information along other axes independent. Therefore, for a d-dimensional tensor with N = S^d, axial attention saves O(N^{(d−1)/d}) computation over standard attention. The proposed axial attention can be used within standard Transformer layers in a straightforward manner to produce Axial Transformer layers, without changing the basic building blocks of traditional Transformer architecture.  The authors did experiments on two standard datasets for generative image and video models: down-sampled ImageNet and BAIR Robot Pushing, and they claim that their proposed method matches or outperforms the state-of-the-art on ImageNet-32 and ImageNet-64 image benchmarks and sets a significant new state-of-the-art on the BAIR Robot Pushing video benchmark.

Reasons to accept:

1.	Simple, easy-to-implement yet effective approach to adapt self-attention to large multidimensional data, which can save considerable computation for efficiency, while still have competitive performance.
2.	Clear writing, with sufficient but not redundant introduction of background knowledge and explanation of both the advantages and drawbacks of existing models (too large computational complexity on high-dimensional data).

Suggestions for improvement:

1.	It would be better if the authors can provide more analysis or case study to show the reason why Axial attention (Axial Transformer) can reach good performance even if it omits considerable operations compared to traditional Transformers, or to show why the attention operations within axis are important instead of attention operations between axis.
2.	Definition of “axis” should be more clear in section 3 (there could be some ambiguities of “axis”).


**Experience Assessment:**

I have read many papers in this area.

**Review Assessment: Checking Correctness Of Derivations And Theory:**

N/A

**Review Assessment: Checking Correctness Of Experiments:**

I assessed the sensibility of the experiments.

**Review Assessment: Thoroughness In Paper Reading:**

I read the paper at least twice and used my best judgement in assessing the paper.

---

### Official Review · AnonReviewer4 · 2019-11-03
**Official Blind Review #4**

**Rating:** 1

**Review:**

It is known that the standard self-attention method is computationally expensive and cost a significantly large amount of storage when the number of points to be attended is large.

This paper attempts to solve this problem and proposed the Axial Attention method. It is claimed to be able to save an O(N^(d-1)/d) factor of resources over standard self-attention.

The proposed method looks novel to me, but some of the related works are missing and the experiment session is insufficient.

1)  The author should at least include the following works which also aim to reduce the cost of self-attention. Since the author did not mention these works which also focus on solving the same problem, It is hard for me to judge if the proposed method is better than existing works.
[a] CCNet: Criss-Cross Attention for Semantic Segmentation
[b] A^2-Nets: Double Attention Networks

2) self-attention has shown its effectiveness on a broad range of computer vision tasks, including image generation, detection, segmentation, and classification. I do not get why the proposed method is only benchmarked for generative models. Is it because the proposed method cannot be adopted on other popular CV tasks, such as detection, segmentation, and classification? Extra experiments should be included if the proposed method is not only designed for generative models.

3) The ablation study is missing. The author directly compared its own method with other existing methods that are implemented and trained with different hyperparameters. It is hard to know which indeed benefits the accuracy gain and how significant is the proposed method.

4) In table 2 and 3, I do not see a clear advantage of the proposed method over the SOTA methods.

**Experience Assessment:**

I have published in this field for several years.

**Review Assessment: Checking Correctness Of Derivations And Theory:**

I assessed the sensibility of the derivations and theory.

**Review Assessment: Checking Correctness Of Experiments:**

I carefully checked the experiments.

**Review Assessment: Thoroughness In Paper Reading:**

I read the paper thoroughly.

---

> ### Author Response · Authors · 2019-11-08
> **Intended scope of paper falls exclusively within generative modelling (here too)**
>
> Thanks for your remarks. Some of your major points (2 and to some extent 1) concern the scope of the notions that we introduce, specifically axial attention. Please note that the intended scope is only axial attention within multidimensional transformers, that is within generative modelling of multidimensional data such as images and videos. We are aiming at making this very clear in the paper. Please see our related remarks to the other reviewers.

---

### Official Review · AnonReviewer5 · 2019-11-03
**Official Blind Review #5**

**Rating:** 3

**Review:**

This paper claims to propose a new approach to solve the computational problems of self-attention. However, the paper mainly focuses on adapting Transformer for image generation, which has far less applications. The whole paper needs to be rewritten to make their target and contribution clearer.

1. The authors overclaim that they provide a new approach for accelerating self-attention. However, they only adapted Transformer for image generation. In fact, Transformer does not equal to self-attention. Currently, two directional self-attention like Bert has much wider applications compared with Transformer like sequential self-attention.

2. For a paper claim to improve self-attention, they should show its effectiveness on a broad range of tasks, with comprehensive experimental evaluation. However, authors mainly reported the image generation on several datasets.

Overall, the authors need to rewrite the paper. They should either show more applications with the proposed self-attention approach or treat it as a new approach for image generation.

**Experience Assessment:**

I have read many papers in this area.

**Review Assessment: Checking Correctness Of Derivations And Theory:**

I assessed the sensibility of the derivations and theory.

**Review Assessment: Checking Correctness Of Experiments:**

I assessed the sensibility of the experiments.

**Review Assessment: Thoroughness In Paper Reading:**

I read the paper at least twice and used my best judgement in assessing the paper.

---

> ### Author Response · Authors · 2019-11-08
> **Intended scope of paper falls exclusively within generative modelling**
>
> Thank you for remarks. Since one of your major objections is at its core the same objection as that by reviewer #1, please see comment above. We want to treat our paper as a new architecture for image (and video) generation and we are making this clear in the text.

---

### Official Review · AnonReviewer1 · 2019-11-08
**Official Blind Review #1**

**Rating:** 1

**Review:**

This paper proposes a novel approach to deal with the computational problems of self-attention without introducing independence assumptions. The proposed approach is simple, easy to understand, and easy to implement.

However, evaluation for this paper is severely lacking. As it is, there is not enough information provided to adequately assess the proposed method's strengths in practice. The following should be added:

Evaluation on a variety of different tasks, such as image segmentation, temporally consistent object detection, object tracking, etc. Why are the evaluations limited to generative modeling? To prove the generality of the method (as claimed), it needs to be applied to various tasks.
Runtime (in inference) comparisons for each of the datasets and for each of the baselines. Additionally, a theoretical analysis for runtime in terms of the size of the input should be given (the column in Table 1 should have runtimes for each method clearly specified, and this should be done for each dataset and baseline)
Ablation study. What is the baseline architecture used without axial attention? There is only comparison to previous work which may have used a different architecture.

If these concerns are thoroughly addressed, I would be happy to increase my score.

**Experience Assessment:**

I have published in this field for several years.

**Review Assessment: Checking Correctness Of Derivations And Theory:**

I assessed the sensibility of the derivations and theory.

**Review Assessment: Checking Correctness Of Experiments:**

I carefully checked the experiments.

**Review Assessment: Thoroughness In Paper Reading:**

I read the paper thoroughly.

---

> ### Author Response · Authors · 2019-11-08
> **Intended scope of paper falls exclusively within generative modelling**
>
> Thank you for your comments. We would like to point outright that the intended scope and focus of the paper is exclusively generative models of images with an extension to videos. Some aspects of the paper make this clear:
> - The title centrally includes "multidimensional transformers" that are only generative models indeed with an encoder part and a decoder part (like the original transformer for language).
> - Our main contribution is the Axial Transformer architecture itself, i.e. how to easily apply (masked) axial attention to multi-dimensional transformers by using a number of additional features: reordering of RGB channels, shifting operation for the rows, shallow and hence faster strict autoregressive decoder, no need for custom kernels.
> - The thorough and exclusive comparison with previous image modelling attention-based architectures.
>
> However, we also realize now that some sentences in the paper may hint at axial attention as a stand-alone operation to be used beyond generative modelling. Showing this is beyond the scope of our paper and we are working to make this clear and rephrase the relevant passages and subsections.

---

> > ### Comment · AnonReviewer1 · 2019-11-08
> > **response**
> >
> > Thank you for your response.
> >
> > Image modeling attention-based architectures is a very narrow scope indeed. If this is truly the scope, I vote to reject the paper. My concerns are as follows:
> >
> > Image modeling is a very broad task. It is currently unclear whether attention-based architectures will be superior to other methods. What is the reason to limit to only this specific subset of image modeling? It seems arbitrary. If image modeling is the true task, a full list of prior work on image modeling should be included and compared with.
> > Transformers are used in many applications beyond image modeling as well. It seems as if the proposed attention mechanism could deliver significant gains in these areas. Is there a reason to focus only on generative image modeling versus other popular CV or NLP tasks as mentioned before? This would be a powerful paper if gains were shown on a wide variety of tasks, with minimal modification to the underlying method (as claimed). As it stands, the scope is too narrow

---

> > > ### Comment · AnonReviewer1 · 2019-11-08
> > > **based on this discussion -- changing rating to reject**
> > >
> > > see title.

---

### Author Response · Authors · 2019-11-15
**Response to all reviewers**

Dear reviewers, thank you for your comments. We have uploaded a revised version of our paper incorporating your feedback. Specifically:

- We are now more explicit about the scope of our paper and its intended contribution. Our work is about autoregressive modeling for images and other data organized as multidimensional tensors -- it falls in the same line of work and scope as papers such as Pixel Recurrent Neural Networks (van den Oord et al 2016), Image Transformer (Parmar et al 2018), Subscale Pixel Networks (Menick and Kalchbrenner 2019), and many others.

- We have included improved results on video modeling (1.29 bits/dim on BAIR robot pushing).

- We have included an ablation study using a baseline architecture for our image model. Specifically, we replace the inner decoder with an LSTM. We find that the results on ImageNet 32x32 and 64x64 are slightly worse by 0.01 and 0.02 bits/dim, respectively, and also that training time is slower than our original model. See Section 4.1 in the revised paper for the full discussion.

- We have included discussion on relationship with other attention proposals in the computer vision literature, such as CCNet. Our contribution and emphasis is on uses of masked axial attention and how to combine it in a way that leads to a valid autoregressive image model (the dependencies between outputs and inputs must obey the raster scan order so that it defines a valid probabilistic model), whereas other works do not employ masking and are not focused on defining an autoregressive model.

- We have also increased our emphasis of the semi-parallel sampling aspect of our model, that is unique among autoregressive image and video models.

All in all we believe the paper, proposed methods, and open source code will be very useful to the generative image modeling community, and we ask you to consider this when making your final decision.

---

### Decision · Program_Chairs · 2019-12-19

**Decision:**

Reject

**Comment:**

This paper proposes a self-attention-based autoregressive model called Axial Transformers for images and other data organized as high dimensional tensors. The Axial Attention is applied within each axis of the data to accelerate the processing.

Most of the authors claim that main idea behind Axial Attention is widely applicable, which can be used in many core vision tasks, such as detection and classification. However, the revision fails to provide more application for Axial attention.

Overall, the idea behind this paper is interesting but more convincing experimental results are needed.